# Rediscovering the *rete ovarii*, a secreting auxiliary structure to the ovary

Dilara N Anbarci[1], Jennifer McKey[1,2], Daniel S Levic[1], Michel Bagnat[1], Blanche Capel[1]*

[1]Department of Cell Biology, Duke University Medical Center, Durham, United States; [2]Section of Developmental Biology, Department of Pediatrics, University of Colorado Anschutz Medical Campus, Aurora, United States

---

### eLife Assessment

This **important** study reports the developmental dynamics and molecular markers of the *rete ovarii* during ovarian development. The data supporting the main conclusions are **convincing**. This study will be of interest to developmental and reproductive biologists.

---

**Abstract** The *rete ovarii* (RO) is an appendage of the ovary that has been given little attention. Although the RO appears in drawings of the ovary in early versions of Gray's Anatomy, it disappeared from recent textbooks, and is often dismissed as a functionless vestige in the adult ovary. Using PAX8 immunostaining and confocal microscopy, we characterized the fetal development of the RO in the context of the mouse ovary. The RO consists of three distinct regions that persist in adult life, the intraovarian rete (IOR), the extraovarian rete (EOR), and the connecting rete (CR). While the cells of the IOR appear to form solid cords within the ovary, the EOR rapidly develops into a convoluted tubular epithelium ending in a distal dilated tip. Cells of the EOR are ciliated and exhibit cellular trafficking capabilities. The CR, connecting the EOR to the IOR, gradually acquires tubular epithelial characteristics by birth. Using microinjections into the distal dilated tip of the EOR, we found that luminal contents flow toward the ovary. Mass spectrometry revealed that the EOR lumen contains secreted proteins potentially important for ovarian function. We show that the cells of the EOR are closely associated with vasculature and macrophages, and are contacted by neuronal projections, consistent with a role as a sensory appendage of the ovary. The direct proximity of the RO to the ovary and its integration with the extraovarian landscape suggest that it plays an important role in ovary development and homeostasis.

---

## Introduction

The *rete ovarii* (RO) is an epithelial structure that is directly connected to the ovary, first described over a century ago (*Waldeyer, 1870*) as a multiregion structure of mesonephric origin. Despite high conservation among mammalian species, including guinea pigs, cows, cats, sheep, swine, prairie deer mice, camels, dogs, monkeys (*Cassali et al., 2000*; *Kim et al., 2012*), cows (*Santos et al., 2012*), and humans (*Khan et al., 1999*), previous work on the RO did not arrive at a consensus on the function of this structure. Thus, it has remained a mysterious orphan structure.

Although the RO has been proposed to be the female homologue of the rete testis (*Wenzel and Odend'hal, 1985*), our recent work revealed that it is more complex (*McKey et al., 2022*). The RO is composed of three distinct regions: (1) the extraovarian rete (EOR), which consists of columnar epithelial cells that create a single convoluted tubule structure ending in a blind distal dilated tip (DDT); (2) the connecting rete (CR), which consists of pseudo-columnar cells; and (3) the intraovarian rete (IOR),

*For correspondence:
blanche.capel@duke.edu

which consists of squamous epithelial cells that branch and form a fine network of thin solid cell cords approximately 1–3 cells thick (**Byskov and Lintern-Moore, 1973**; **McKey et al., 2022**). It was previously shown that the RO consists of ciliated and non-ciliated cells rich in apical microvilli and mitochondria (**Czernobilsky et al., 1985**). Fine granular Periodic acid-Schiff (PAS)-positive material was found in the cytoplasm of the EOR and CR cells, indicating the presence of polysaccharides (e.g., glycogen and mucins) (**Stein and Anderson, 1979**). Because the proportion of PAS-positive material in the RO was found to be influenced by the estrous cycle in cows (**Wenzel et al., 1987**), researchers concluded that the luminal contents of the RO were under endocrine control (**Archbald et al., 1971**). Although these results strongly suggested a secretory role for the RO, this function was not experimentally confirmed, and no proteomic or metabolomic investigations were pursued. Previous characterization of the RO relied heavily on imaging serial sections to study the structure. However, these studies lacked contextual information, and the use of sections made it challenging to determine whether tubules were connected to one another or were isolated structures (**Wenzel and Odend'hal, 1985**; **Woolnough et al., 2000**). Furthermore, some investigators reported that the whole RO contracted postnatally and that the DDT of the EOR separated from the RO and degenerated (**Byskov and Lintern-Moore, 1973**). Perhaps for these reasons, the RO was deemed a functionless vestige and has been omitted from recent textbook representations of the female reproductive tract (**Girsh, 2021**).

The RO was rediscovered and highlighted in our recent study (**McKey et al., 2022**), where we used confocal and lightsheet imaging of whole ovaries to study the integration of ovary morphogenesis with the development of surrounding tissues, including the RO. Recently, the IOR has gained attention as a newly described progenitor for supporting cells of the murine gonad (**Mayère et al., 2022**). This echoes previous data suggesting that the RO plays a role in the onset of meiosis (**Stein and Anderson, 1979**). Recent single-cell transcriptomics studies have also identified cells of the RO within human fetal gonads (**Lardenois et al., 2023**; **Taelman et al., 2022**). We reported that the entire RO expresses high levels of PAX8 and used this as a marker to visualize and characterize cells of the RO. In the present study, we used the Pax*8-rtTa; Tre-H2B-GFP* RO nuclear reporter mouse line, advanced imaging techniques, and secretome analysis to characterize the development of the intact RO in its native context and to investigate the function and heterogeneity of its cells. We found that the RO arises from a subset of the mesonephric tubules, analogous to the rete testis, and persists into adulthood. Our studies reveal that the RO is a continuous structure, surrounded by smooth muscle actin, a dense vascular network, and several macrophage populations. We also show that the RO is directly contacted by neurons. The enrichment of secretory machinery in RO cells as well as our experimental analysis of directional flow and luminal contents, together suggest that the RO sends material to the ovary. Based on these findings, we suggest that the RO plays a role in ovary function and should be investigated as a functional organ of the female reproductive tract.

## Results

### Characterization of the IOR, CR, and EOR

We used immunofluorescence (IF) and confocal imaging to investigate the development and subregional structure of the RO from embryonic day (E) 16.5 to postnatal day (P) 7. Three distinct regions of the RO were originally defined histologically by cell morphology (**Byskov and Lintern-Moore, 1973**; **Lee et al., 2011**). Using IF and confocal imaging, we found that these three regions are maintained throughout development, but their relative sizes change. While the entirety of the RO is PAX8+, we took different approaches to identify region specific markers. First, as GFRa1 was known to be expressed in the rete testis, we hypothesized that it was expressed in the RO as well. Indeed, immunostaining using antibodies against GFRa1 revealed that part of the RO is GFRa1+. GFRa1 specifically labeled the CR from E16.5 to P7 (**Figure 1**, bottom row). To identify additional RO markers, we performed bulk and single-cell RNA sequencing (ScRNA-seq) of cells from the ovarian complex of mice at E16.5 and 2 months and found that *Krt8* was enriched in the RO (**McKey et al., 2022**; **Anbarci et al., 2024**). We validated these findings using IF against KRT8, which revealed that KRT8+ cells were specifically localized to the EOR (**Figure 1**, third row; **McKey et al., 2022**).

To characterize RO development, we next analyzed PAX8 and KRT8 expression from E16.5 to P0. At E16.5, the CR and IOR were the largest of the three regions. In the EOR, several tubules leading from the region of the CR converged into a single tube of columnar PAX8+ and KRT8+ epithelial cells that

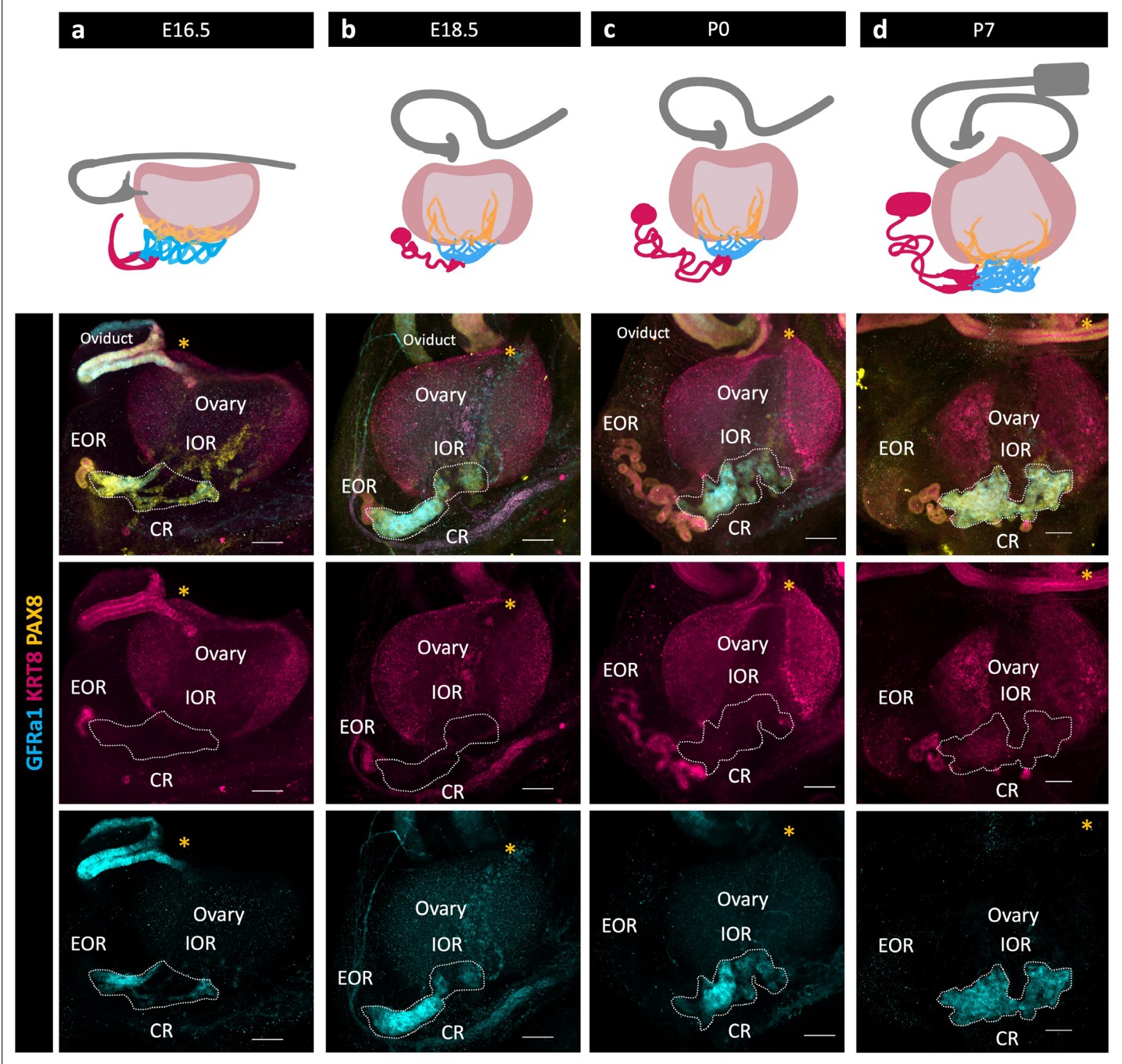

**Figure 1.** The *rete ovarii* (RO) undergoes dynamic changes during development alongside the ovary. (**a–d**, top row) Diagrams depicting the development of the three regions (extraovarian rete [ER], magenta; connecting rete [CR], cyan; intraovarian rete [IOR], yellow) of the RO from E16.5 to P7 in the whole ovarian complex (ovary, light pink; oviduct, gray). (**a–d**, bottom rows) Maximum intensity projection from confocal Z-stacks of whole ovary/ mesonephros complexes at E16.5 (**a**), E18.5 (**b**), P0 (**c**), and P7 (**d**) immunostained for PAX8 (yellow), GFRa1 (cyan), and KRT8 (magenta). (**a–d**, second row) are composite images, while the third row shows separate panels for KRT8 and the bottom row shows panels for GFRa1. *Note that GFRa1 and KRT8 do not co-localize and are specific to the CR and EOR, respectively. All figures are dorsal views of the ovary. Yellow asterisk indicates opening of the infundibulum for reference. Scale bar – 100 um.

The online version of this article includes the following figure supplement(s) for figure 1:

**Figure supplement 1.** Ventral view of ovary and extraovarian rete (EOR) during development.

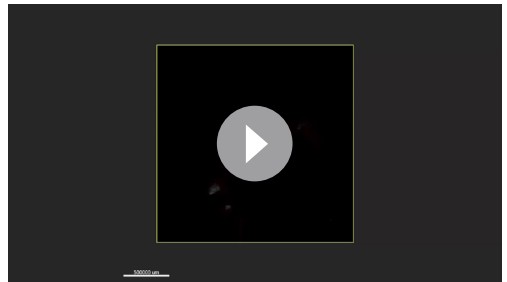

**Video 1.** 3D model of an XX ovarian complex at E18.5. This video depicts the 3D rendering of native lightsheet images of an E18.5 ovary/mesonephros complex. It demonstrates how the rete wraps around the ovary. DNA (gray) ovarian surface epithelium (LGR5; green) and PAX8 (red).

https://elifesciences.org/articles/96662/figures#video1

lead to a blind end. Between E16.5 and E18.5, the EOR underwent rapid expansion, and the blind end of the EOR became dilated. This structure, which we refer to as the DDT, was best visualized from the ventral side of the ovary (*Figure 1— figure supplement 1*, *Video 1*). By E18.5, the EOR was the largest of the three regions. At this stage, the CR was still large, but the IOR began to regress. At P0, the EOR remained the largest region, and the IOR had regressed to the medullary region of the ovary (*McKey et al., 2022*). Their distinct structure and protein expression patterns suggest that each region has a different function.

## Integration of the EOR with the extraovarian environment

During gonadogenesis, the mesonephros is highly vascularized in both XX and XY embryos (*Cool et al., 2011*). In contrast, the ovary and surrounding tissue are more highly innervated than the testis (*McKey et al., 2019*). A surprising finding in our bulk transcriptome analysis of the RO was the presence of a high proportion of immune cells and cells with vascular markers that co-isolated with E16.5 RO cells (*Anbarci et al., 2024*). To explore the integration of the RO with its environment, we used IF to investigate the expression of endothelial marker Endomucin (*Figure 2a*), smooth muscle marker alpha smooth muscle actin (aSMA) (*Figure 2b*),

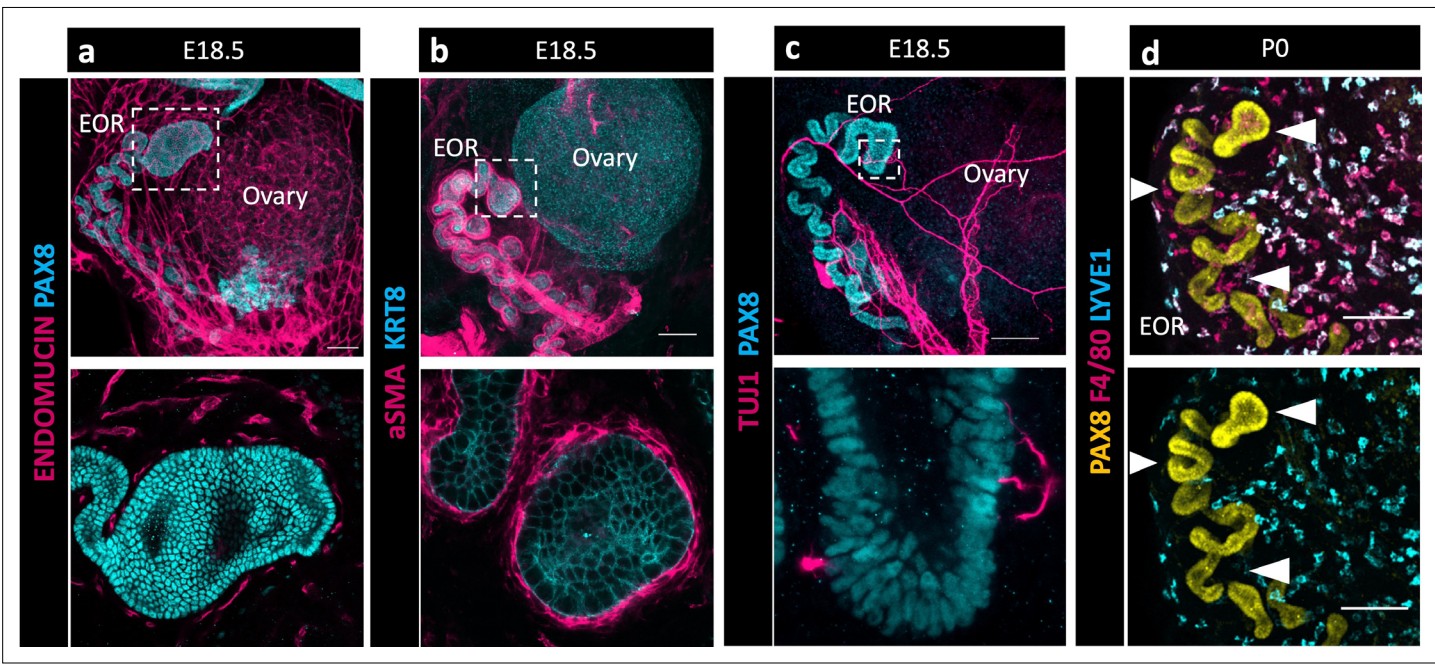

**Figure 2.** The extraovarian rete (EOR) is highly integrated with its extraovarian environment suggesting multifaceted communication. (**a**) Top panel is a whole ovarian complex maximum intensity projection of the confocal Z-stack at E18.5 immunostained for ENDOMUCIN (magenta) and PAX8 (cyan). Bottom panel is an optical section showing vasculature tightly surrounding the distal dilated tip (DDT) of the EOR (magenta). (**b**) Top panel is a whole ovarian complex maximum intensity projection of the confocal Z-stack at E18.5 immunostained for aSMA (magenta) and KRT8 (cyan). Bottom panel is an optical section showing the EOR tightly ensheathed by smooth muscle (magenta). (**c**) Top panel is a whole ovarian complex maximum intensity projection of the confocal Z-stack at E18.5 immunostained for TUJ1 (magenta) and PAX8 (cyan). Bottom panel is an optical section showing direct contacts between the EOR and neuronal projections (magenta). (**d**) EOR at P0 immunostained for PAX8 (yellow), F4/80 (magenta), and LYVE-1 (cyan). Top and bottom panels are maximum intensity projection of the confocal Z-stack. Bottom panel shows the absence of LYVE1 macrophages (cyan) proximate to the EOR, in contrast to the closely associated F4/80 macrophages (magenta in top image). Arrowheads show regions devoid of LYVE1 macrophages where F4/80 macrophages are present. Scale bar – 100 um.

pan-neuronal marker TUJ1 (*Figure 2c*), and macrophage markers F4/80 and LYVE1 (*Figure 2d*). We found that at E18.5 the EOR was tightly surrounded by vasculature and ensheathed within a layer of aSMA+ mesenchyme. The EOR, and more specifically the DDT, was directly contacted by neurons that contact the PAX8+ epithelial cells. We also found that the EOR is specifically associated with F4/80+ macrophages (*Figure 2d*). This multifaceted integration with the environment suggests the RO may respond to or interpret homeostatic cues.

## Flow of luminal material within the EOR

Because the EOR lumen is fluid filled, and because the EOR and CR cells are rich in proteoglycans (*Byskov, 1975*), we hypothesized that the EOR produces luminal secretions. To investigate this idea, we first chose to determine the direction of flow at P7 when the DDT of the EOR was fully dilated (*Figure 1d*). We first injected fluorescently labeled pH-insensitive dextran into the DDT of the EOR and found that, within just 15 minutes, the fluorescent fluid had readily traveled from the DDT into the ovary, where it then diffused widely (*Figure 3*). By contrast, when we injected fluorescently labeled dextran into the P7 ovary near the IOR, the dextran remained in the ovary and did not travel to the EOR (*Figure 3*). These data indicate that, at least at P7, the fluid inside the lumen of the EOR travels toward the ovary.

## Potential of the EOR and CR for fluid transfer

Without the obvious indication of a lumen within the CR at P7 and taking into consideration the absence of the epithelial marker KRT8 in those cells, we wondered how fluid could travel through the CR to the ovary. Using an antibody against E-CADHERIN, a marker of cell junctions, we found that it was specifically present in the EOR at E16.5, but absent from the CR (*Figure 4a*, second row). However, E-CADHERIN expression was gradually gained in the CR and IOR such that, by P7, the entire RO was positive for this epithelial marker (*Figure 4a–d*, second row). These results suggest that tubular epithelial connections between the EOR and IOR mature gradually between E16.5 and P7.

Next, we investigated mechanisms that could facilitate fluid movement through the RO. Previous reports showed that ciliated and non-ciliated cells are present in the RO (*Lee et al., 2011*). Using an antibody against a marker for cilia, ARL13b, we found that primary ciliated cells were abundant in the DDT and throughout the tubules of the EOR at E18.5 (*Figure 4g and h*). Due to the curvature of the DDT, optical slices often include cilia from neighboring cells. However, an ultrathin (0.6 um) optical section suggested that each cell has a single cilium. Since the EOR was covered in a sheath of aSMA+ mesenchyme (*Figures 2b and 4e and f*), we investigated whether and when this layer of smooth muscle became contractile. We used an antibody against the contractile smooth muscle protein Calponin (CNN1) and found that CNN1 was absent around the EOR during fetal development and in neonates. However, by P7, the mesenchymal sheath around the EOR gained expression of CNN1, indicating that it acquired the ability to contract by this stage of development (*Figure 4e and f*). These data suggest that ciliary mechanosensing and/or muscle contraction may aid in the directional movement of the fluid from the DDT to the ovary.

## Proteins produced by the EOR indicate a role for the SNARE-complex

Our data showing fluid flow from the EOR to the ovary prompted us to investigate the nature and identity of the proteins produced and potentially secreted by the EOR. To address this question, we analyzed the protein contents of the luminal material using mass spectrometry. Using the *Pax8rtTA; Tre-H2B-GFP* reporter mouse line, we dissected EORs and isolated luminal fluid by gently pressing the tissue with a pestle. Using this method, we anticipated several problems. First, we expected that some cells within the EOR and surrounding tissue would be lysed during this procedure, which would release proteins not ordinarily secreted. Second, we expected that some cells closely associated with the EOR would be co-isolated, which could result in contamination with contents from cells that are not part of the EOR. To eliminate intracellular proteins arising from lysed cells, we cross-referenced our proteomic dataset with that of the mammalian secretome (*Meinken et al., 2015*), thereby retaining only known secreted proteins. Next, we compared the resulting list to the E16.5 ScRNA-seq data from cells mapping to the RO (*Anbarci et al., 2024*). This produced a conservative candidate list of 15 proteins (*Table 1*). Of the candidate proteins, two were selected for validation, *CLU* and *STXBP2*,

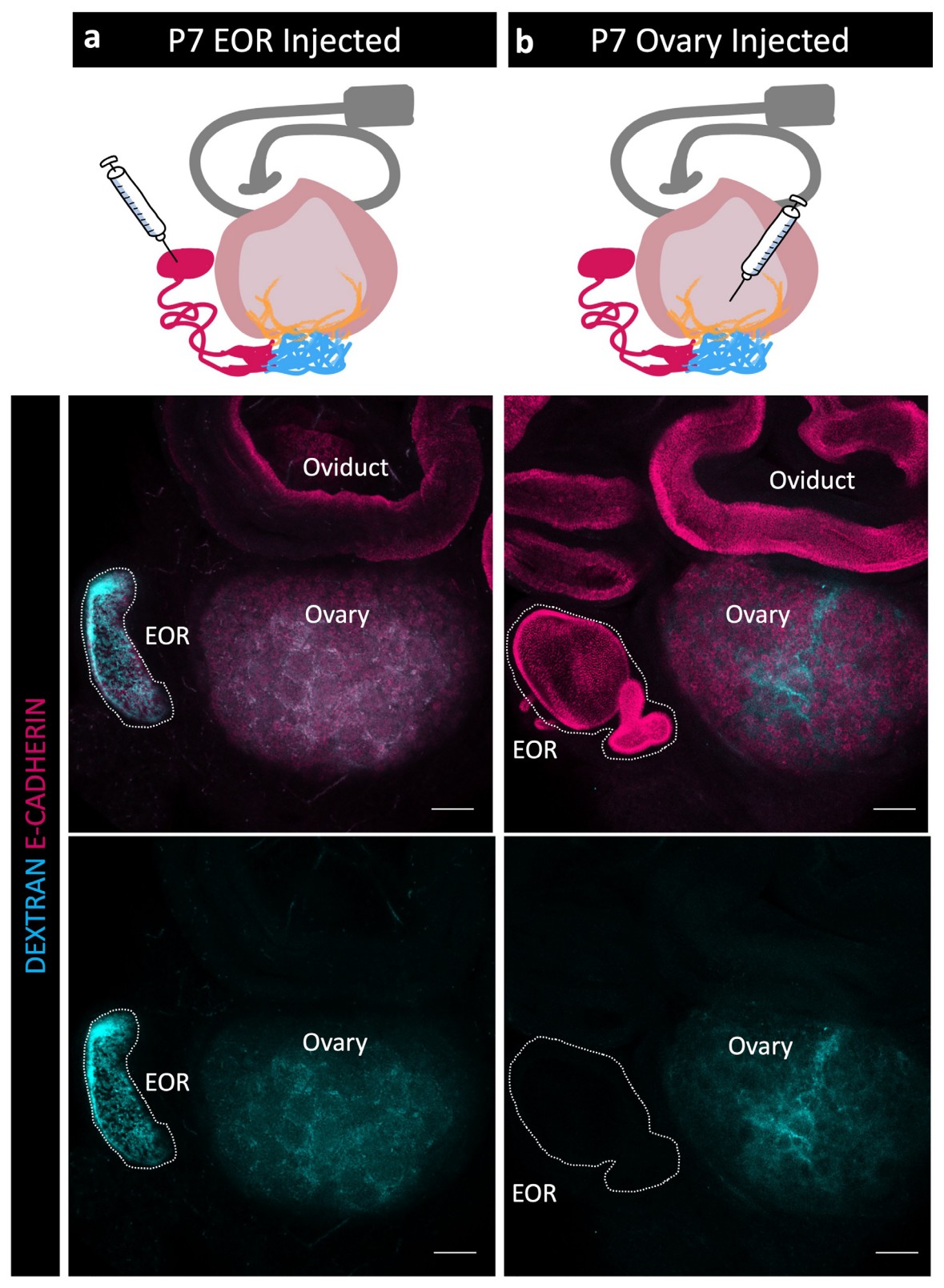

**Figure 3.** Fluid moves from the extraovarian rete (EOR) to the ovary. (**a, b**) Schematic of dextran injections (syringe) for each group (**a**), EOR injections; (**b**,), ovary injections. Oviduct, gray; ovary, light pink; intraovarian rete (IOR), yellow; connecting rete (CR), cyan; EOR, magenta. (Second row) Maximum intensity projection of a confocal Z-stack of whole ovarian complexes at P7, where dextran was injected into the EOR. Presence of dextran in the ovary shows that when dextran is injected into the EOR, it diffuses throughout the ovary (dextran, cyan; E-CADHERIN, magenta). (Bottom row) Maximum

*Figure 3 continued on next page*

*Figure 3 continued*

intensity projection from confocal Z-stacks of whole ovary/mesonephros complexes at P7 where dextran was injected into the ovary. Absence of dextran signal in the EOR shows that dextran did not travel into the EOR when injected into the ovary near the IOR. Scale bar – 100 um.

due to their roles in protein and vesicle transport (*Argraves and Morales, 2004*; *Saewu et al., 2017*; *Söllner, 2003*).

Because no validated antibodies were commercially available to visualize protein expression for CLU and STXBP2, validations were performed using hybridization chain reaction (HCR), a method for single-molecule RNA-fluorescence in situ hybridization (FISH) (https://files.molecularinstruments.com/MI-Protocol-RNAFISH-Mouse-Rev9.pdf). We found that both *Clu* and *Stxbp2* were expressed in the EOR at E18.5 (*Figure 5a and b*) and at P7 (*Figure 5c and d*). The presence of *Stxbp2* showed that components of the SNARE-complex were actively transcribed in the EOR, suggesting that this may serve as a mechanism for secretion. The presence of other components of the SNARE-complex were validated using IF. We found that the T-SNARE complex member STX3 was expressed throughout the EOR and was localized to the apical surface, a cellular position that is typically associated with active

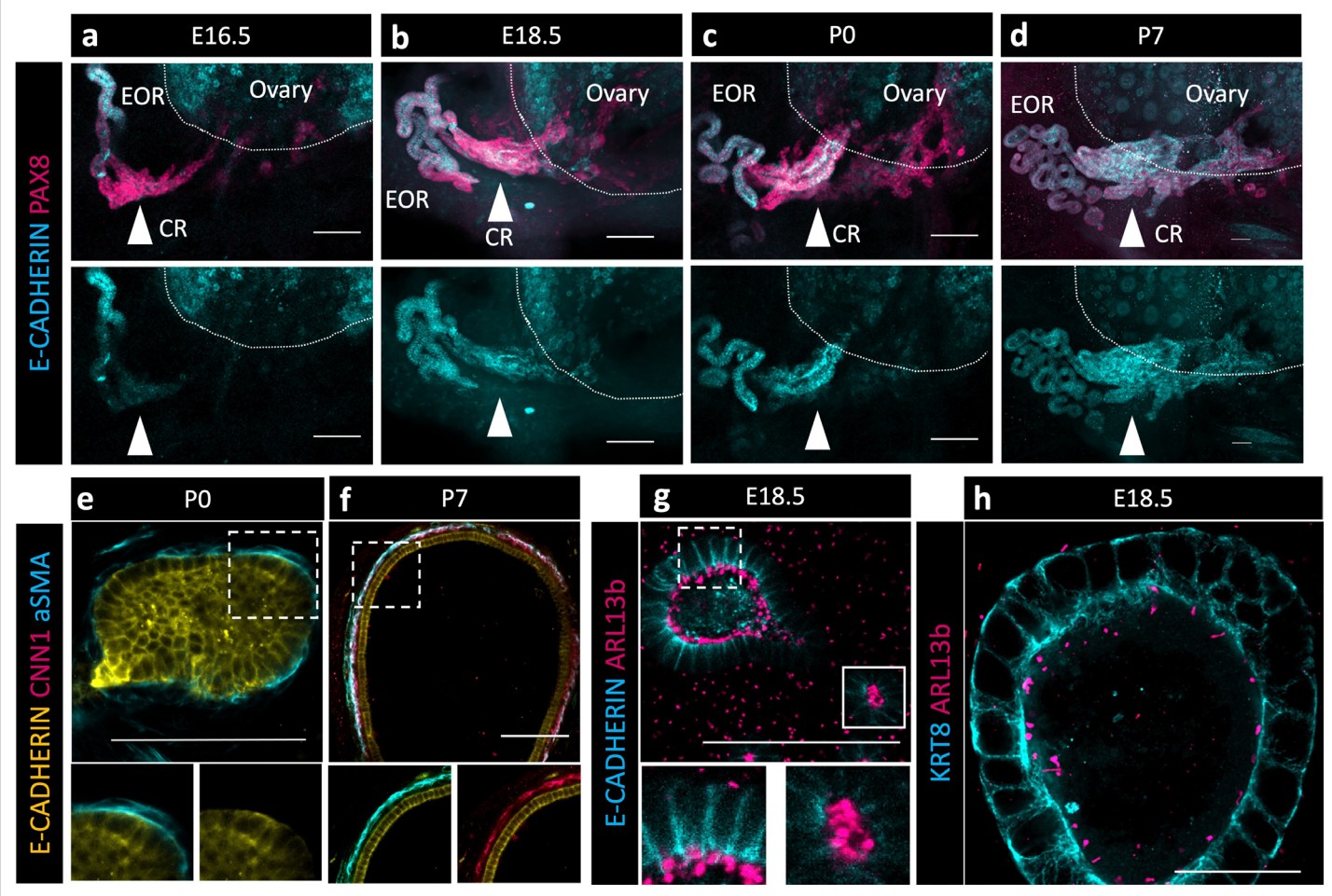

**Figure 4.** The extraovarian rete (EOR) and connecting rete (CR) acquire the potential for fluid transfer. (**a–d**) Cells of the CR acquire E-CADHERIN soon after birth. Maximum intensity projection from confocal Z-stacks of whole ovary/mesonephros complexes at E16.5-P7 immunostained for PAX8 (magenta) and E-CADHERIN (cyan). Arrowhead shows the region of the CR where E-CADHERIN expression is negligible at E16.5 (**a**), low at E18.5 (**b**), and fully expressed at P0 (**c**) and P7 (**d**). (**e**) Optical section from confocal Z-stacks of RO complexes at P0 immunostained for E-CADHERIN (yellow), CNN1 (magenta), and aSMA (cyan). Boxed regions show the absence of CNN1 in the aSMA+ sheath around the EOR at P0, and (**f**) its acquisition by P7, suggesting gain of contractility. (**g**) The RO expresses cilia marker ARL13b by P0. Maximum intensity projection from confocal Z-stacks of RO complexes at E18.5 immunostained for E-CADHERIN (cyan) and ARL13b (magenta). Outlined boxes show all regions of the EOR have ciliated cells. Scale bar – 100 um. (**h**) Airyscan optical section from confocal Z-stacks of RO complexes at E18.5 immunostained for KRT8 (cyan) and ARL13b (magenta). Scale bar – 25 um.

**eLife** Research article

Cell Biology | Developmental Biology

**Table 1.** Cells of the extraovarian rete (EOR) secrete proteins essential for vesicle transport.

Candidate list of 15 proteins identified by mass spectrometry. Asterisked proteins were selected for validation due to role in vesicle and protein transport.

| Protein | Role |
| --- | --- |
| Agt | Pro-peptide for angiotensinogen |
| Bcam | Glycoprotein |
| C3 | Complement component |
| Cd200 | Glycoprotein |
| Cfi | Serine proteinase |
| Clu* | Secreted chaperone |
| Cp | Metalloprotein |
| Cpm | Membrane-bound arginine/lysine carboxypeptidase |
| Epcam | Antigen |
| Igfbp2 | Binds insulin-like growth factors I and II |
| Lama5 | Laminin |
| Napsa | Pro-peptide |
| Sema3c | Secreted glycoprotein |
| Slit3 | Secreted protein |
| Stxbp2* | Binds syntaxin |

secretion (*Figure 5e*; *Söllner, 2003*). Ras-associated binding (RAB) proteins are required in the SNARE complex to tether vesicles to the T-SNARE and allow fusion and secretion (*Takahashi et al., 2012*). Using an antibody against the small GTPase found on the surface of vesicles, RAB11, we found that it was also localized to the sub-apical region of EOR cells (*Figure 5f*), which is the typical docking position for vesicles prior to exocytosis (*Söllner, 2003*). Taken together, these data suggest that the EOR actively secretes proteins into the lumen of the structure possibly encapsulated in extracellular vesicles.

## Proteins produced by the EOR suggest a role in ovary homeostasis

Among proteins captured in our mass spectrometry analysis of the EOR, insulin-like growth factor binding protein 2 (IGFBP2) stood out as a secreted protein with a potential functional role in ovary homeostasis. IGFBP2, found to be secreted by granulosa cells, binds and sequesters IGF1 (*Rosenzweig, 2004*). The binding of IGFBP2 to IGF1 titrates IGF1 from its receptor IGF1R (*Amutha and Rajkumar, 2017*). IGF1 has a reported role in ovarian function by amplifying the hormonal action of gonadotropins to promote steroidogenesis and granulosa cell proliferation (*Talia et al., 2021*).

Because no validated antibodies were commercially available to visualize protein expression for IGFBP2, we again used HCR to determine when and where *Igfbp2* was expressed in the epithelial cells of the RO. We found that *Igfbp2* was highly expressed in the EOR at E18.5 (*Figure 6a*). The mRNA was still present, but at lower levels in P7 EOR cells (*Figure 6b*).

## Discussion

The female reproductive tract is often thought to consist of the vulva, vagina, cervix, uterus, oviduct, and ovaries (*Rendi et al., 2012*). We suggest that the RO be added to this list and investigated as an additional component of female reproductive function. We show that the RO consists of three distinct regions that mature during fetal life and persist into adulthood. The cells of the EOR show secretory capabilities consistent with the presence of secreted proteins in the lumen identified through mass spectrometry. Importantly, we show that contents of the EOR are transported to the ovary by P7, suggesting a role in ovary homeostasis.

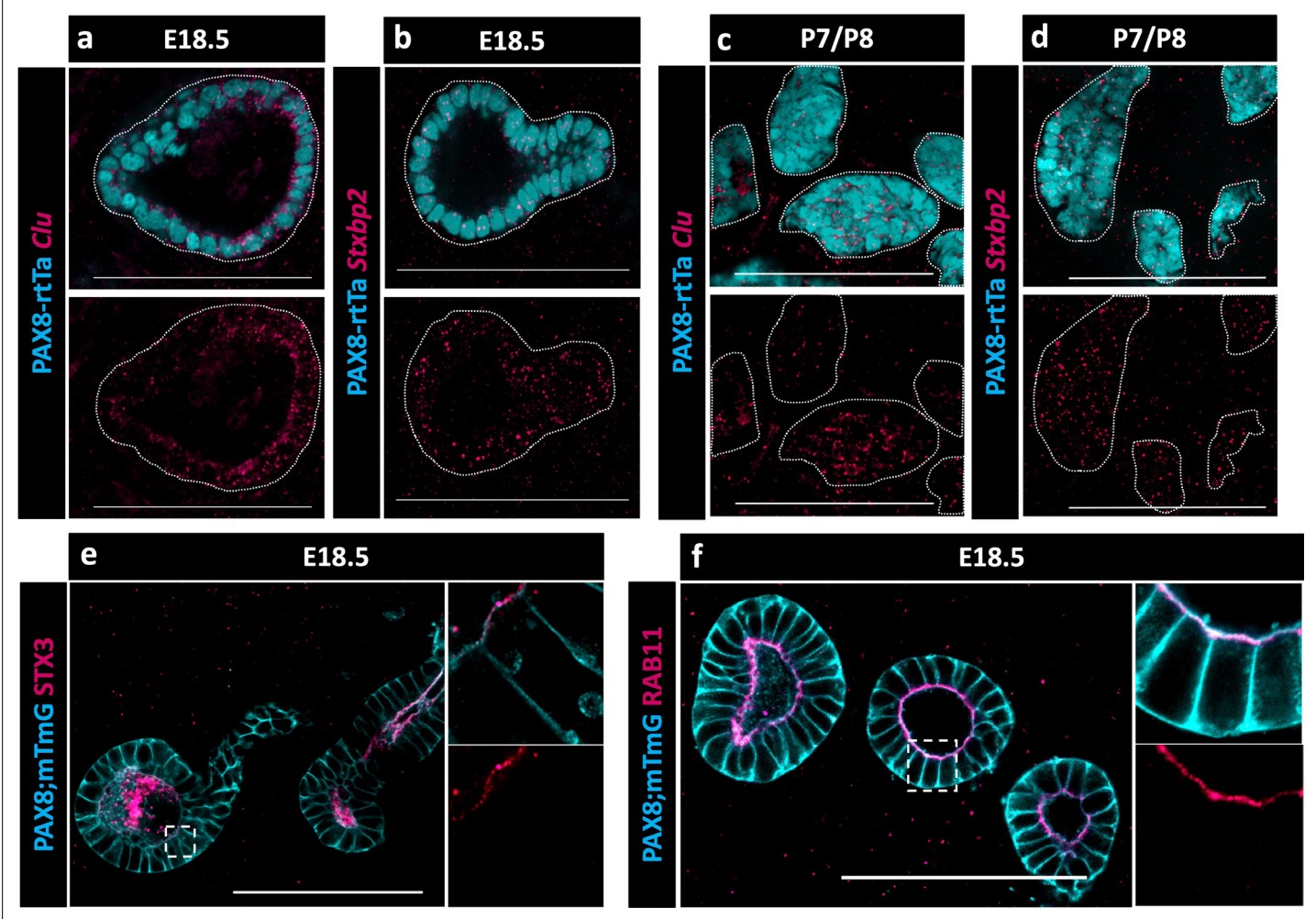

**Figure 5.** Presence of SNARE-complex members suggests a role for secretion. (**a**) Optical section from confocal Z-stacks of PAX8-rtTa; Tre-H2B-GFP (cyan) extraovarian rete (EOR) at E18.5 using hybridization chain reaction (HCR) for the detection of Clu expression (magenta). (**b**) Optical section from confocal Z-stacks of PAX8-rtTa; Tre-H2B-GFP (cyan) EOR at E18.5 using HCR for Stxbp2 (magenta). (**c**) Optical section from confocal Z-stacks of PAX8-rtTa; Tre-H2B-GFP (cyan) EOR at P7/8 using HCR for Clu (magenta). (**d**) Optical section from confocal Z-stacks of PAX8-rtTa; Tre-H2B-GFP (cyan) EOR at P7/8 using HCR for Stxbp2 (magenta). (**e**) Optical section from confocal Z-stacks of PAX8-rtTa; Tre-Cre; Rosa26mTmG (cyan) EOR at E18.5 immunostained for STX3 (magenta). Outlined higher resolution image acquired with Airyscan. (**f**) Optical section from confocal Z-stacks of PAX8-rtTa; Tre-Cre; Rosa26mTmG (cyan) EOR at E18.5 immunostained for RAB11 (magenta). Outlined higher resolution image acquired with Airyscan. Scale bar – 100 um.

Historically, the regions of the RO were identified using histological sections, where the EOR was defined as a columnar epithelium, while the CR was defined as pseudo-columnar. Recently, we found that the entire RO expresses PAX8+, which is usually considered a marker of urogenital epithelial identity. We also showed that the IOR contained PAX8+/FOXL2+ cells and that the EOR was KRT8+ at E17.5 (*McKey et al., 2022*). In this study, we further characterized KRT8 and E-Cadherin expression throughout RO development as the CR acquires epithelial characteristics. We also identified a marker, GFRa1, that is exclusively expressed in the CR. These molecular distinctions predict that the different regions of the RO are functionally distinct.

While PAS-positive staining in the lumen of the EOR of sheep indicated the presence of glycogens, glycoproteins, and proteoglycans (*Stein and Anderson, 1979*), it remained unclear whether these were secreted by the EOR cells or were accumulated secretions from the ovary. Using injections of a membrane-impermeable dye, dextran, into either the RO or ovary, we found that the fluid within the EOR travels toward the ovary. We also found that the EOR is ensheathed within a layer of aSMA+ mesenchyme that expresses the contractile component Calponin (CNN1) (*Gao et al., 2014*). Our

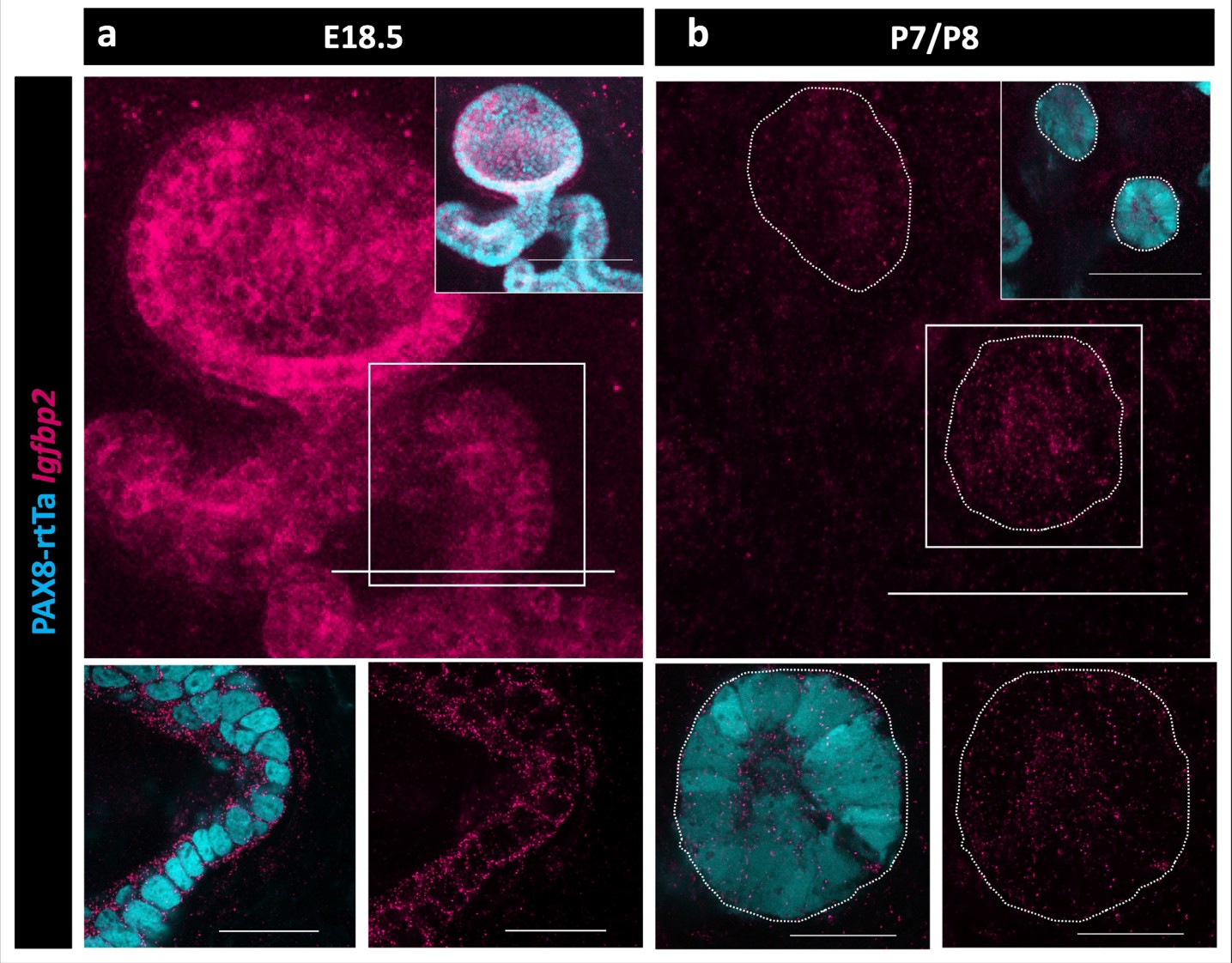

**Figure 6.** Validation of Igfbp2 expression in the *rete ovarii* (RO). (**a**) Maximum intensity projection of confocal Z-stacks of PAX8-rtTa; Tre-H2B-GFP (cyan) extraovarian rete (EOR) at E18.5 using hybridization chain reaction (HCR) for the detection of IGFBP2 expression (magenta). (**b**) Maximum intensity projection of confocal Z-stacks of PAX8-rtTa; Tre-H2B-GFP (cyan) EOR at P7/8 using HCR for the detection of IGFBP2 expression (magenta). Bottom panels are outlined. Higher resolution images acquired with Airyscan. Scale bar top panel – 100 um. Scale bar bottom panel – 25 um.

data suggests that the EOR gains contractility during development, which could facilitate fluid movement within the EOR lumen in conjunction with cilia within the EOR. We found that the ciliary protein ARL13b was expressed in the EOR. While the primary cilium is not motile and does not generate flow, we found in our ScRNA-seq data that cells in the EOR express *Pkd2*. In renal epithelial cells, nonmotile primary cilia expressing PDK2 sense shear stress during fluid flow and transduce this sensory information (*Nauli et al., 2003*). Perhaps fluid movement within the EOR is a two-part process, where the primary cilia sense pressure within the lumen inducing contraction of the smooth muscle surrounding the EOR to promote fluid movement toward the ovary. Our E16.5 ScRNA-seq data indicates that the EOR only expresses *Arl13b* and does not express the multicilia marker *Foxj1*. However, both markers are present in the putative EOR cell clusters in our adult SnRNA-seq data (*Anbarci et al., 2024*), suggesting that, after maturation, the EOR likely includes *Arl13b+/Foxj1+* multiciliated cells. It is well documented that adult oviductal multiciliated cells express both ARL13b and FOXJ1 (*Coy et al., 2016*). It is possible that the primary cilia give rise to motile multiciliated cells, as shown in the airway epithelium (*Jain et al., 2010*) and suggested in the oviduct (*Shi et al., 2014*).

We used a 'milking' process to extrude the contents of EOR, and mass spectrometry, to identify 4232 proteins. We expected this group to be a mix of proteins from the luminal fluid, and cytoplasmic proteins from lysed EOR cells or from cells closely associated with the EOR. We took a very conservative approach to identify proteins highly likely to be secreted from EOR cells: (1) we compared the proteomics data with the known mammalian secretome to identify only secreted proteins; and (2) we next compared this shortened list with our available E16.5 ScRNA-seq data to find proteins whose transcripts were specific to the EOR at that stage. Although we recognize that the secretome at P7 is likely far more complex, this candidate list of 15 proteins reveals the first insight into specific proteins secreted from the RO that may be involved in the secretory process itself or may be transported to the ovary to affect ovary function.

Within the protein candidate list, we focused on validating clusterin (CLU) and syntaxin binding protein 2 (STXBP2) due to their roles in protein and vesicle transport. Consistent with our proteomic results, both *Clu* and *Stxbp2* RNAs were detected in the RO. CLU is a chaperone protein that is present in both reproductive and non-reproductive tissues (*Argraves and Morales, 2004*; *Saewu et al., 2017*). In the epididymis, CLU acts as a chaperone to direct luminal proteins to the sperm head surface (*Saewu et al., 2017*). Although the role of CLU in the RO remains unknown, we hypothesize that it is important for post-secretory protein guidance to the ovary. STXBP2 binds t-SNARE protein STX and is essential for the vesicle-apical surface membrane fusion stage of vesicle secretion (*Söllner, 2003*). The presence of STXBP2 in the candidate list, coupled with validation of the expression of *Stxbp2* RNA in the cells of the EOR, led us to investigate the presence of other components of the SNARE complex in the EOR.

The SNARE complex is comprised of a vesicle bound v-SNARE (Vamps), membrane bound t-SNARE (STXs), small GTPases (RABs), and syntaxin binding proteins (STXBPs) (*Söllner, 2003*). Using IF staining, we confirmed the presence of STX3, as well as RAB11. Our ScRNA-seq data indicate that EOR cells also express *Vamp7* and *Vamp8*, which are v-SNAREs typically associated with STX3 (*Dingjan et al., 2018*). Small GTPases, such as RAB11, are essential in the SNARE complex as they mediate the tethering of vesicle membranes to the apical membrane prior to membrane fusion. Taking into consideration the presence of all components of the SNARE complex, we hypothesize that the EOR utilizes the SNARE complex to promote apical secretion into the luminal fluid.

A protein that was prominent on the proteomic candidate list was IGFBP2. We found that *Igfbp2* was highly expressed in both the embryonic and adult RO transcriptome (*Anbarci et al., 2024*). IGFBP2 is an ideal candidate protein to mediate ovarian function. The IGF1 pathway is essential for the regulation of follicular growth and selection (*Talia et al., 2021*). IGFBP2 can bind and sequester IGF1 (*Rosenzweig, 2004*), limiting its availability to promote follicle growth through activation of its receptor IGF1R (*Amutha and Rajkumar, 2017*; *Bezerra et al., 2018*; *Liu et al., 2021*). We spatially validated the expression of *Igfbp2* in the EOR using HCR and noted high expression at E18.5, which was decreased by P7. This decrease correlated with the end of the first wave of follicle activation immediately after birth (*Mork et al., 2012*; *Zheng et al., 2014*). We also found variation in levels of *Igfbp2* in our adult SnRNA-seq data, where *Igfpb2* was enriched specifically in RO cells during the estrus stage compared to the diestrus stage of the estrous cycle. Based on these results, we hypothesize that high levels of estradiol during estrus trigger the EOR to produce IGFBP2 to sequester free IGF1 and decrease follicle growth and production of estradiol, maintaining ovary homeostasis.

The EOR is highly integrated with the extraovarian environment. We found that EOR cells are tightly surrounded in a dense web of vasculature. The RO may send or receive information through the vasculature, similar to other epithelial tissues in the female reproductive tract (*Reynolds et al., 2002*), raising the possibility that the RO participates in endocrine signaling. We also found that the EOR is directly contacted by TUJ1+ neurons, another avenue through which information may be entering or exiting the EOR. It is unclear whether the innervation contacts the epithelial cells specifically, similar to the way in which innervation contacts epithelial cells in the gut (*Bohórquez et al., 2015*), or whether the innervation contacts the smooth muscle cells that surround the EOR to induce contraction, similar to the innervation-muscle interaction in the uterus (*Morizaki et al., 1989*). We also found that the EOR is specifically associated with F4/80+LYVE1- macrophages. LYVE1 macrophages are predicted to have an angiogenic role in the adipose tissue surrounding the epididymis and could possibly be playing a similar role in the EOR (*Cho et al., 2007*). The integration of the EOR with the extraovarian environment, including vasculature, neurons, and macrophages, places the EOR in an ideal position

to act as an antenna to interpret homeostatic signals and send information to the ovary. In *Drosophila* and *Caenorhabditis elegans*, there is ample evidence that the ovary responds to physiologic levels of glycogen, insulin, amino acid levels, and changes in diet (*Hubbard et al., 2013*; *Laws and Drummond Barbosa, 2017*). The mediator of this response has not been determined in mammals, but we hypothesize that this is the function of the RO. Interestingly, our scRNAseq data revealed expression of ESR1, PGR, INSR, and IGF1R, which are all critical hormone receptors regulating different aspects of female reproduction and health. Ongoing work will investigate how the EOR responds to hormones and other physiological signals, and whether proteins secreted by the EOR such as IGFBP2 respond to physiologic stimuli such as diet or immune status, and convey this information to the ovary.

## Materials and methods

**Key resources table**

| Reagent type (species) or resource | Designation | Source or reference | Identifiers | Additional information |
|---|---|---|---|---|
| Strain, strain background (*Mus musculus*) | Crl:CD1(ICR) | Charles River | Strain code: 022; RRID:IMSR_CRL:022 | |
| Strain, strain background (*M. musculus*) | C57BL/6J | Jackson Laboratory | Stock #:000664; RRID:IMSR_JAX:000664 | |
| Genetic reagent (*M. musculus*) | Tre-H2B-Gfp (Tg(tetO-HIST1H2BJ/GFP)47Efu/J) | PMID:14671312 | MGI:J:90563; RRID:IMSR_JAX:005104 | |
| Genetic reagent (*M. musculus*) | Tre-Cre (B6.Cg-Tg(tetO-cre)1Jaw/J) | PMID:12145322 | MGI:J:78365; RRID:IMSR_JAX:006234 | |
| Genetic reagent (*M. musculus*) | mTmG Gt(ROSA)26Sortm4(ACTB-tdTomato,-EGFP)Luo/J | PMID:17868096 | MGI:J:124702; RRID:IMSR_JAX:007576 | |
| Genetic reagent (*M. musculus*) | Pax8-rtTA (B6.Cg-Tg(Pax8-rtTA2S*M2)1Koes/J) | PMID:18724376 | MGI:J:140925; RRID:IMSR_JAX:007176 | |
| Genetic reagent (*M. musculus*) | Lgr5 (B6.129P2-Lgr5tm1(cre/ERT2)Cle/J) | PMID:17934449 | MGI:J:127123; RRID:IMSR_JAX:008875 | |
| Antibody | Smooth muscle alpha action (aSMA) (Cy3-conjugated mouse monoclonal) | Sigma-Aldrich | C6198; RRID:AB_476856 | (1:1000) |
| Antibody | E-Cadherin (rat monoclonal) | Zymed (Thermo Fisher Scientific) | 13-1900; RRID:AB_2533005 | (1:500) |
| Antibody | Endomucin (rat monoclonal) | Santa Cruz Biotechnology | sc-65495; RRID:AB_2100037 | (1:500) |
| Antibody | GFP (chicken polyclonal) | Abcam | ab13970; RRID:AB_300798 | (1:1000) |
| Antibody | GFRa1 (goat polyclonal) | R&D Systems | AF560; RRID:AB_2110307 | (1:150) |
| Antibody | KRT8 (rat monoclonal) | DSHB | TROMA-I; RRID:AB_531826 | (1:250) |
| Antibody | PAX8 (rabbit polyclonal) | Proteintech | A10336-1-AP; RRID:AB_2918972 | (1:500) |
| Antibody | CNN1 (rabbit polyclonal) | Proteintech | 13938-1-AP; RRID:AB_2082010 | (1:200) |
| Antibody | ARL13b (rabbit polyclonal) | Proteintech | 17711-1-AP; RRID:AB_2060867 | (1:1000) |
| Antibody | RAB11 (rabbit monoclonal) | Cell Signaling Technology | 5589; RRID:AB_10693925 | (1:500) |

*Continued on next page*

*Continued*

| Reagent type (species) or resource | Designation | Source or reference | Identifiers | Additional information |
|---|---|---|---|---|
| Antibody | LYVE-1 (goat polyclonal) | R&D Systems | AF2125; RRID:AB_2297188 | (1:500) |
| Antibody | F4/80 (rat monoclonal) | Bio-Rad | MCA497RT; RRID:AB_1102558 | (1:2000) |
| Antibody | STX3 (rabbit monoclonal) | Abcam | ab133750 | (1:200) |
| Antibody | TUJ1 (488-conjugated mouse monoclonal) | BioLegend | A488-435L; RRID:AB_10143904 | (1:1000) |
| Antibody | AF647 anti-Rabbit (donkey polyclonal) | Jackson ImmunoResearch | 711-605-152; RRID:AB_2492288 | (1:500) |
| Antibody | AF488 anti-Chicken (donkey polyclonal) | Jackson ImmunoResearch | 703-545-155; RRID:AB_2340375 | (1:500) |
| Antibody | AF488 anti-Rat (donkey polyclonal) | Life Technologies | A-21208; RRID:AB_2535794 | (1:500) |
| Antibody | Cy3 anti-Goat (donkey polyclonal) | Jackson ImmunoResearch | 705-165-147; RRID:AB_2307351 | (1:500) |
| Sequence-based reagent | Stxbp2 (B3 amplifier) | Molecular Instruments | Accession #: XR_001778418.1 | |
| Sequence-based reagent | Clu (B1 amplifier) | Molecular Instruments | Accession #: NM_013492.3 | |
| Sequence-based reagent | IGFBP2 (B3 amplifier) | Molecular Instruments | Accession #: NM_008342.3 | |
| Sequence-based reagent | F647 (B3 amplifier) | Molecular Instruments | | |
| Sequence-based reagent | F546 (B1 amplifier) | Molecular Instruments | | |
| Chemical compound, drug | Dextran, Alexa Fluor 568; 10,000 MW | Thermo Fisher | D22912 | |
| Chemical compound, drug | Dichloromethane | MilliporeSigma | 270997-1L | |
| Chemical compound, drug | Benzyl Ether | MilliporeSigma | 108014-1KG | |
| Chemical compound, drug | Quadrol=N,N,N',N'-Tetrakis(2-Hydroxypropyl)ethylenediamine | MilliporeSigma | 122262 | |
| Software, algorithm | Zen Black Edition | Carl Zeiss | | |
| Software, algorithm | Imaris v9.6 | Bitplane | | |
| Software, algorithm | Adobe Creative Cloud | Adobe | | Photoshop, Illustrator, Premier Pro |

## Mice and tissue collection

Unless otherwise stated, mice used for experiments were maintained on the CD-1 or mixed CD-1 and C57BL/6J genetic backgrounds. The *Pax8-rtTA* and *Tre-H2B-GFP* lines were previously described (*Traykova-Brauch et al., 2008*; *Tumbar et al., 2004*) and maintained on a mixed CD-1/C57BL/6J background, and maintained on a mixed CD-1/C57BL/6J background. The *Pax8-rtTa;Tre-Cre;Rosa26^{mTmG}* line was obtained by crossing the *Pax8-rtTa* line with carriers of the *Tre-Cre* (*Perl et al., 2002*) and *Rosa26^{mTmG} (mTmG)* (*Muzumdar et al., 2007*) alleles, which allowed for visualization of cell membranes of PAX8+ cells. Pregnant and nursing dams were given a doxycycline diet at 625 mg/kg (Teklad Envigo TD.01306) 3 days prior to tissue collection to induce GFP expression in Pax8+ cells. Toe samples were collected from mice for genotyping. The primers used for PCR genotyping are listed in *Supplementary file 1*. To obtain samples at specific stages of development, males were housed with females for timed matings. Successful mating was determined by the presence of a vaginal plug. Date of plug was considered embryonic day 0.5. Tissue samples were collected in phosphate-buffered saline

(PBS) without calcium or magnesium, fixed in 4% paraformaldehyde (PFA)/PBS for 30 minutes at room temperature and dehydrated stepwise into 100% methanol, followed by storage at –20°C. All mice were housed in accordance with National Institutes of Health guidelines, and experiments were conducted with the approval of the Duke University Medical Center Institutional Animal Care and Use Committee (protocol #: A089-20-04 9N).

## Immunostaining and confocal image acquisition

Samples were stepwise rehydrated into 100% PBS, followed by a 30-minute permeabilization wash in PBS 0.1% Triton X-100. Samples were then blocked for 1 hour with PBS, 1% Triton X-100, 10% horse serum, and 3% BSA. Samples were incubated overnight in primary antibodies diluted in blocking solution at 4°C (Key Resource Table). Samples then underwent three 30-minute washes in permeabilization solution and incubated overnight in secondary antibodies (1:500 dilution) and Hoechst vital dye solution diluted in blocking solution. The next day, samples underwent three 20-minute washes in permeabilization solution before being transferred into 100% PBS and stored at 4°C until ready for confocal imaging. Samples were mounted in DABCO mounting solution and stored at –20°C until imaged. Samples were imaged from both the dorsal and ventral sides using 3D-printed reversible slides that utilize two coverslips that allow for flipping. The 3D model can be downloaded on the NIH 3D Print Exchange website at https://3dprint.nih.gov/discover/3DPX-009765. Samples were imaged in toto using laser scanning confocal microscopy captures in the longitudinal plane on Zeiss LSM780 or LSM880 and affiliated Zen software (Carl Zeiss, Inc, Germany) using ×10, ×20, and ×63 (also used for Airyscan) objectives.

## Hybridization chain reaction

The mouse embryo protocol from Molecular Instruments whole-mount (available here) was adjusted for P0 and P7 ovary samples. Samples were stepwise rehydrated into 0.1% Tween 20/PBS (PBST). Samples were then subjected to 10 μg/ml proteinase K solution for 10 minutes at room temperature, then washed twice in PBST for 5 minutes. Samples were then post-fixed in 4% PFA for 10 minutes at room temperature, followed by three PBST washes for 5 minutes. Samples were then pre-hybridized for 30 minutes in 500 ul of hybridization buffer (Molecular Instruments) at 37°C. Samples were then incubated overnight at 37°C with HCR probes diluted in 500 ul hybridization buffer (2 pmol in 500 ul). Samples were then washed for 15 minutes four times in the Molecular Instruments probe wash buffer at 37°C, followed by two 5 × 0.1% Tween 20/SSC (SSCT) washes for 5 minutes at room temperature. Fluorescently labeled hairpins were snap cooled and left in a dark drawer at room temperature for 30 minutes. Following hairpin preparation, samples were incubated with hairpins diluted in Molecular Instruments amplification buffer overnight in the dark at room temperature. On the third day, samples were washed in SSCT four times for 15 minutes each before being transferred to PBS and then mounted in DABCO mounting medium for imaging.

## Image processing

Confocal images were imported into FIJI software for minor image processing (cropping, rotations, maximum intensity projection/optical slice montage, and color application). The RO was oriented to the left of the ovary with the oviduct at the top of the ovary. Images were then imported into Adobe Photoshop CC (Adobe, Inc, CA) for final processing of channel overlay, brightness, contrast adjustment, and modification of red to 'magenta' (hue adjustment to –30). Channel color accessibility was determined via Photoshop colorblind proofing.

## Dextran injections

Postnatal day 7 mice were euthanized, and the entire ovarian complex was dissected in PBS with magnesium and calcium. Samples were then placed on agar blocks (1.5%) soaked in PBS with magnesium and calcium. The distal tip of the EOR was identified and punctured using a tungsten needle (0.001 mm tip diameter, Fine Science Tools, 10130-05). A microinjection unit (Picospritzer III micro-injector) and fine capillary glass needle (5–10 um) loaded with 2–3 nl dextran solution (1.25 mg/ml dextran, PBS, 0.05% phenol red) was used to inject dextran into the opening of the EOR. Samples were then left for 15 minutes before fixing with 4% PFA. During the 15 minutes, samples were monitored to visualize movement. Samples were then fixed, stained, and imaged.

## Luminal fluid collection for proteomics and analysis

Postnatal day 7 EORs were collected from *Pax8-rtTa; Tre-H2B-GFP* mice (as a guide to only collect EOR) and carefully cleaned up to remove as much non-EOR tissue (40 EOR samples). Samples were then placed in a 1.5 ml Eppendorf tube and 'pressed' for fluid using a disposable pellet pestle. The sample was then spun down and the supernatant was collected and snap frozen in liquid nitrogen. The sample was then submitted to the Duke University Proteomics Core for mass spectrometry. Proteomic analysis uncovered 4252 proteins present in the pressed fluid. To exclude proteins that may have contributed due to cell lysis, results were cross-referenced against a database of secreted proteins (*Meinken et al., 2015*). Overlapping proteins were then cross-referenced to our E16.5 ScRNA-seq data (*Anbarci et al., 2024*), and candidate proteins were determined by gene expression specific to the EOR cluster.

## LC-MS/MS proteomics analysis

The sample was subjected to a Bradford (Pierce) protein measurement and 10 ug was removed for downstream processing. The sample was brought to 4% SDS, reduced with 10 mM dithiothreitol for 20 min at 55°C, alkylated with 25 mM iodoacetamide for 45 min at room temperature, and then subjected to S-trap (Protifi) trypsin digestion using manufacturer-recommended protocols. Digested peptides were lyophilized to dryness and resuspended in 50 ul of 0.2% formic acid/2% acetonitrile. The sample was subjected to chromatographic separation on a Waters MClass UPLC equipped with a 1.8 μm Acquity HSS T3 C18 75 μm × 250 mm column (Waters Corp.) with a 90 min linear gradient of 5–30% acetonitrile with 0.1% formic acid at a flow rate of 400 nl/minute with a column temperature of 55°C. Data collection on the Fusion Lumos mass spectrometer with a FAIMS Pro device was performed for three difference compensation voltages (–40v, –60v, –80v). Within each CV, a data-dependent acquisition mode of acquisition with a $r$ = 120,000 (@ m/z 200) full MS scan from m/z 375–1500 with a target AGC value of 4e5 ions was performed. MS/MS scans with HCD settings of 30% were acquired in the linear ion trap in 'rapid' mode with a target AGC value of 1e4 and max fill time of 35 ms. The total cycle time for each CV was 0.66 s, with total cycle times of 2 s between like full MS scans. A 20 s dynamic exclusion was employed to increase depth of coverage. The total analysis cycle time for each sample injection was approximately 2 hours.

Raw LC-MS/MS data files were processed in Proteome Discoverer 3.0 (Thermo Scientific) and then submitted to independent Sequest database searches against a *Mus musculus* protein database containing both forward and reverse entries of each protein. Search tolerances were 2 ppm for precursor ions and 0.8 Da for product ions using trypsin specificity with up to two missed cleavages. Carbamidomethylation (+57.0214 Da on C) was set as a fixed modification, whereas oxidation (+15.9949 Da on M) was considered a dynamic mass modifications. All searched spectra were imported into Scaffold (v5.3, Proteome Software) and scoring thresholds were set to achieve a peptide false discovery rate of 1% using the PeptideProphet algorithm. Data were output as total spectral matches.

## Acknowledgements

The authors would like to thank Benjamin Carlson and Lisa Cameron from the Duke Light Microscopy Core Facility for confocal imaging resources and assistance on high-resolution imaging and analysis. The authors would also like to thank Dr. Erik Soderblom from the Duke Center for Genomic and Computational Biology core facility for proteomic analysis and assistance. The authors would also like to thank Vidita Shah for technical assistance. We are grateful to all members of the Capel and Bagnat laboratories for their constant support, discussion, and suggestions on the work presented in this paper and beyond. This project was supported by grants from the National Institutes of Health #1R01HD090050-0 to BC, #1R01DK132120 to MB, and #K99HD103778 and #R00HD103778 to JM. DNA was supported by #5R01HD039963 to BC and the Duke School of Medicine.

## Additional information

### Competing interests

Michel Bagnat: Reviewing editor, *eLife*. The other authors declare that no competing interests exist.

## Funding

| Funder | Grant reference number | Author |
|---|---|---|
| National Institutes of Health | #1R01HD090050-0 | Blanche Capel |
| National Institutes of Health | #1R01DK132120 | Michel Bagnat |
| National Institutes of Health | K99HD103778 | Jennifer McKey |
| National Institutes of Health | #R00HD103778 | Jennifer McKey |
| National Institutes of Health | #5R01HD039963 | Dilara N Anbarci<br>Blanche Capel |
| Duke University | Trent Fund | Blanche Capel |

The funders had no role in study design, data collection and interpretation, or the decision to submit the work for publication.

## Author contributions

Dilara N Anbarci, Conceptualization, Data curation, Formal analysis, Validation, Investigation, Visualization, Methodology, Writing - original draft, Writing – review and editing; Jennifer McKey, Conceptualization, Resources, Investigation, Visualization, Methodology, Writing – review and editing; Daniel S Levic, Investigation, Methodology, Writing – review and editing; Michel Bagnat, Resources, Funding acquisition, Methodology, Writing – review and editing; Blanche Capel, Conceptualization, Resources, Supervision, Funding acquisition, Project administration, Writing – review and editing

## Author ORCIDs

Dilara N Anbarci  https://orcid.org/0000-0003-0435-4749
Jennifer McKey  https://orcid.org/0000-0002-2640-1502
Daniel S Levic  https://orcid.org/0000-0003-0240-5178
Michel Bagnat  https://orcid.org/0000-0002-3829-0168
Blanche Capel  https://orcid.org/0000-0002-6587-0969

## Ethics

All mice were housed in accordance with National Institutes of Health guidelines, and experiments were conducted with the approval of the Duke University Medical Center Institutional Animal Care and Use Committee (protocol #: A089-20-04 9N).

Reviewer #1 (Public review): https://doi.org/10.7554/eLife.96662.3.sa1
Reviewer #2 (Public review): https://doi.org/10.7554/eLife.96662.3.sa2
Reviewer #3 (Public review): https://doi.org/10.7554/eLife.96662.3.sa3
Author response https://doi.org/10.7554/eLife.96662.3.sa4

# Additional files

## Supplementary files
MDAR checklist

Supplementary file 1. PCR Genotyping Primer Sequences.

## Data availability

All data generated or analyzed during this study are included in the manuscript and/or supplementary materials.

The following previously published dataset was used:

| Author(s) | Year | Dataset title | Dataset URL | Database and Identifier |
|---|---|---|---|---|
| Anbarci DN, O'Rourke R, Xiang Y, Peters DT, Capel B, McKey J | 2024 | Transcriptome analysis of the mouse fetal and adult rete ovarii and surrounding tissues | https://www.ncbi.nlm.nih.gov/geo/query/acc.cgi?acc=GSE244849 | NCBI Gene Expression Omnibus, GSE244849 |

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
