## [Editor Report · eLife Assessment]

This **important** study reports the developmental dynamics and molecular markers of the *rete ovarii* during ovarian development. The data supporting the main conclusions are **convincing**. This study will be of interest to developmental and reproductive biologists.

---

## [Referee Report · Reviewer #1 (Public review)]

Summary:

The manuscript by Anbarcia et al. re-evaluates the function of the enigmatic Rete Ovarii (RO), a structure that forms in close association with the mammalian ovary. The RO has generally been considered a functionless structure in the adult ovary. This manuscript follows up on a previous study from the lab that analyzed ovarian morphogenesis using high-resolution microscopy (McKey et al., 2022). The present study adds finer details to RO development and possible function by (1) identifying new markers for OR sub-regions (e.g. GFR1a labels the connecting rete) suggesting that the sub-regions are functionally distinct, (2) showing that the OR sub-regions are connected by a luminal system that allows transport of material from the extra-ovarian rete (EOR) to the inter-ovarian rete (IOG), (3) identifies proteins that are secreted into the OR lumen and that may regulate ovarian homeostasis, and finally, (4) better defines how the vasculature, nervous, and immune system integrates with the OR.

Strengths:

The data is beautifully present and convincing. They show that the RO is composed of three distinct domains that have unique gene expression signatures and thus likely are functionally distinct.

---

## [Referee Report · Reviewer #2 (Public review)]

A large number of ovarian experiments have been conducted - especially in morphological and molecular biology studies - specifically removing the ovarian membrane. This experiment is a good supplement to existing knowledge and plays an important role in early ovarian development and the regulation of ovarian homeostasis during the estrous cycle. There are also innovations in research ideas and methods, which will meet the requirements of experimental design and provide inspiration for other researchers.

Comments on revisions: I don't have any further opinions and suggest to accept.

---

## [Referee Report · Reviewer #3 (Public review)]

Summary:

The rete ovarii (RO) has long been disregarded as a non-functional structure within the ovary. In their study, Anbarci and colleagues have delineated the markers and developmental dynamics of three distinct regions of the RO - the intraovarian rete (IOR), the extraovarian rete (EOR), and the connecting rete (CR). Notably focusing on the EOR, the authors presented evidence illustrating that the EOR forms a convoluted tubular structure culminating in a dilated tip. Intriguingly, microinjections into this tip revealed luminal flow towards the ovary containing potentially secreted functional proteins. Additionally, the EOR cells exhibit associations with vasculature, macrophages, and neuronal projections, proposing the notion that the RO may play a functional role in ovarian development during critical ovariogenesis stages. By identifying marker genes within the RO, the authors have also suggested that the RO could serve as a potential structure linking the ovary with the neuronal system.

Strengths:

Overall, the reviewer commends the authors for their systematic research on the RO, shedding light on this overlooked structure in developing ovaries. Furthermore, the authors have proposed a series of hypotheses that are both captivating and scientifically significant, with the potential to reshape our understanding of ovarian development through future investigations.

Weaknesses:

Although the manuscript lacks conclusive data to support many of its conclusions, the authors provide highly constructive discussions that offer valuable insights for future research on the rete ovarii in the field.

---

## [Author Response]

The following is the authors’ response to the original reviews.

**Reviewer #1 (Public review):**
Weaknesses:It is not always clear what the novel findings are that this manuscript is presenting. It appears to be largely similar to the analysis done by McKey et al. (2022) but with more time points and molecular markers. The novelty of the present study's findings needs to be better articulated.

The previous study focused on placing the Rete Ovarii in the context of ovarian development. The current study focuses on the novel findings that the EOR is a active structure that sends fluid/information to the ovary. We show this by characterizing the presence of secretory proteins in the RO epithelial cells, by dye injections into the EOR and observing transport of the dye to the ovary, and by collection of EOR fluid followed by proteomic analysis. We also show that RO is embedded in an elaborate vascular network and contacted by neurons. None of this data was not discussed in the McKey 2022 paper.

**Reviewer #2 (Public Review):**
Clarifications:(1) Is there any comparative data on the proteomics of RO and rete testis in early development? With some molecular markers also derived from rete testis, it would be better to provide the data or references.

To the best of our knowledge, there are no available proteomic datasets of the embryonic or early postnatal mouse Rete Testis or Epididymis. The authors agree that having this information would be very useful.

(2) Although the size of RO and its components is quite small and difficult to operate, the researchers in this article had already been able to perform intracavitary injection of EOR and extract EOR or CR for mass spectrometry analysis. Therefore, can EOR, CR, or IOR be damaged or removed, providing further strong evidence of ovarian development function?

We attempted to genetically ablate the RO by expressing the diphtheria toxin receptor (DTR) in RO cells and adding DT. This approach was not successful in ablating the RO. We also tried to use Pax2/8 homo- and heterozygous mutants for ablation (as used in the McKey 2022 paper), but so far, we cannot find a genetic combination that ablates the RO, but not the oviduct, uterus and/or kidneys. We have also embarked on a study to surgically remove the RO. This assay is taking some time to optimize. The goal of the current study was to characterize the cells along the length of the RO and to present evidence that it is a secretory appendage of the ovary.

(3) Although IOR is shown on the schematic diagram, it cannot be observed in the immunohistochemistry pictures in Figure 1 and Figure 3. The authors should provide a detailed explanation.

An annotation has been added to Figure 1 to indicate the IOR. As the images within the panels are of maximum intensity projections, it is often difficult to clearly see the IOR as it is deeper within the ovary. In Figure 3, the view of the ovary is from the ventral side: this view does not allow for clear visualization of the IOR.

**Reviewer #3 (Public Review):**
Weaknesses:There is a lack of conclusive data supporting many conclusions in the manuscript. Therefore, the paper's overall conclusions should be moderated until functional validations are conducted.

We have moderated the conclusions where appropriate

**Reviewer #1 (Recommendations For The Authors):**
(1) The introduction is relatively brief and does not mention some historical data/hypotheses on the role of the RO in ovarian function (e.g. regulation of meiotic entry) or development (e.g. Mayère et al., 2022).

Mayere 2022 was cited in line 57. Steins hypothesis about entry into meiosis has been added line 58.

(2) L82-84: It is stated that KRT8 was first identified as a potential RO marker by sc/snRNAseq (Anbarci et al., 2023) and then validated in this manuscript. However, KRT8 was used by McKey et al. (2022) as a RO marker, and they noted there that KRT8 was enriched in the EOR. It is not clear why McKey et al. is not cited as the primary reference validating KRT8 as an EOR marker.

The embryonic and neonatal timecourse description from KRT8 expression is first identified in this paper. McKey 2022 only highlights KRT8 at E18.5 A reference has been added to address this line 85

(3) Figure 1: Can the IOR be seen in these images? If so, please label.

The label has been added.

(4) L107: It is hypothesized that "the RO may respond to or interpret homeostatic cues." Can transcriptomics data shed light on what signals the RO may be capable of responding to? E.g. what receptors are expressed by cells of the RO (e.g. ER, LHCGR, FSHR)?

The RO expresses ESR1, PGR, INSR, IGF1R. The IOR exclusively expresses LHCGR and FSHR.This has been added to the manuscript line 309

(5) L152: Mass spec was used to identify proteins secreted into the lumen of the RO. These proteins were then compared to the mammalian secretome to filter out possible nonsecreted protein contaminants. Finally, the candidates were compared to the RO scRNAseq data from Anbarci et al., (2023). This method gives a very conservative candidate list. However, it may also be informative to compare the sc/snRNA-seq gene list directly to the secretome to ID other possible candidate-secreted proteins that may not have been detected in the mass spec data set.

There are quite a number of secreted proteins that are also not actively secreted. This is a good suggestion for future analysis. For the current study we wanted to take a more conservative approach, and chose to do proteomics to determine proteins that are actively secreted.

(6) L195: It is not clear if IGFBP2 is expressed by both OR and granulosa cells or only granulosa cells. It would be informative to know what ovarian cell types express both IGFBP2 and IGF1R (e.g. from sc/snRNA-seq)? This information is referenced in the discussion (L285-287) but would be better to reference it in the results section for clarity.

Both RO and granulosa cells express IGFBP2 and IGF1R. A sentence has been added to results for clarity. (Line 197)

(7) L295: "...the RO participates in endocrine signaling..." might be more accurate to say "...the RO responds to endocrine signaling...".

The authors agreed that this statement is more accurate and the changes have been made.

**Reviewer #3 (Recommendations For The Authors):**
Several issues significantly affect the paper's quality in the current version. Firstly, there is a lack of conclusive data supporting many conclusions in the manuscript. For instance, the assertion in line 105 that "EOR was directly innervated by neurons" lacks substantial evidence beyond basic immunofluorescent staining.

We agree that the term “innervated” might be a step too far since we rely on IF evidence. We changed the wording of this sentence to say, “The EOR was directly contacted by neurons”.

In another pivotal experiment illustrated in Figure 3, the provided images lack temporal continuity and quantitative analysis, suggesting the incorporation of time-lapse imaging for improved sequential presentation in Figure 3.

The microscope where we can perform injections cannot record movies. We have tried moving the rete to another microscope after injection, but so far, we have been unable to capture dextran moving through the RO. We therefore believe that transport is rapid, but future experiments will be needed to optimize this imaging.

Moreover, relying solely on proteomics analysis, as seen in lines 188-189, makes it challenging to assert conclusions such as "EOR actively secretes proteins." Therefore, the paper's overall conclusions should be moderated until functional validations are conducted.

The findings that (1) the cells of the EOR express SNARE complex proteins at their apical surfaces and (2) luminal fluid expelled from the EOR contains abundant secreted proteins strongly suggest that the RO is involved in active secretion. We use the word “suggest” in this sentence, lines 188-189 as we realize that further experiments should be done to validate this conclusion.

Furthermore, the predominant methods in this study involve immunostaining and imaging. However, the current images exhibit a notable inconsistency in color definitions for different markers by the authors. For instance, in Figure 2.A/C, PAX8 is portrayed as cyan, while in D, it is represented in yellow. Similarly, in Figure 4, E-CAD is depicted using both cyan and yellow. Utilizing different colors for the same protein within a figure can significantly confuse readers' interpretation of the experiments. Rectifying these inconsistencies is essential to enhance the clarity and comprehension of the experimental results.

These colors were chosen to be visible to those with color image impairments. We typically used cyan and magenta to emphasize the most important markers in the image. When E-Cad and KRT8 were often used to emphasized or landmark a structure by localization of these protein. When KRT8 and E-Cad were highlighted, they were represented in cyan and magenta for visibility. When these proteins were used as a landmark to orient the reader and not as the main point, they were labeled in yellow.

At last, many markers in this study are derived from bulk and single-cell sequencing of developing RO. However, it seems that these important data were separated into another paper as a preprint. If this data were incorporated into the current manuscript, the manuscript would become more comprehensive for guiding future research on the RO.

Since we have single cell and single nuclei data from fetal and adult estrus and metestrus stages, we found that incorporating all this data into the present manuscript was overwhelming. Instead, we devoted another manuscript to presenting and validating that data. We believe a quick look at the sequencing manuscript will make this clear.